# Effect of ZrO₂ Content on Microstructure Evolution and Sintering Properties of (Tb$_{0.7}$Lu$_{0.3}$)$_2$O$_3$ Magneto-Optic Transparent Ceramics

**Yu Xin** [1,2]**, Tao Xu** [1]**, Yaozhi Wang** [1,2]**, Peng Luo** [1]**, Weiwei Li** [2]**, Bin Kang** [1]**, Bingchu Mei** [2,*] **and Wei Jing** [1,*]

1   Institute of Chemical Materials, China Academy of Engineering Physics, Mianyang 621900, China
2   State Key Laboratory of Advanced Technology for Materials Synthesis and Processing, Wuhan University of Technology, Wuhan 430070, China
*   Correspondence: bcmei@whut.edu.cn (B.M.); jingwei@caep.cn (W.J.)

**Abstract:** In this paper, (Tb$_{0.7}$Lu$_{0.3}$)$_2$O$_3$ magneto-optical transparent ceramics with different ZrO$_2$ doping levels (0~5 at%) were prepared by hydrogen sintering and sequential HIP technique using ZrO$_2$ as a sintering aid. The effect of ZrO$_2$ doping content on the microstructure and optical properties of (Tb$_{0.7}$Lu$_{0.3}$)$_2$O$_3$ ceramics was analyzed. We found that the optimal doping content of ZrO$_2$ was 3 at%. The transmittance of 3 at% ZrO$_2$-doped (Tb$_{0.7}$Lu$_{0.3}$)$_2$O$_3$ ceramics at the wavelength of 1064 nm was 74.84 %, and the Verdet constant was approximately 275.28 rad·T$^{-1}$·m$^{-1}$ at the wavelength of 650 nm.

**Keywords:** magneto-optic ceramics; Tb$_2$O$_3$; Lu$_2$O$_3$; ZrO$_2$; microstructure; sintering properties

## 1. Introduction

A magneto-optical element (MOE) is one of the crucial components of Faraday rotators and isolators and can eliminate reflected light to ensure the stability of laser transmission in the laser system [1–4]. Usually, magneto-optical materials with high Verdet constants are chose for MOEs because small magneto-optical materials are required if the Verdet constant high according to the Faraday effect formula of $\theta = VBL$, where $V$ is the Verdet constant, $\theta$ is the rotation angle of the light vector and $L$ is the length of the magneto-optical material.

At present, a TGG (Terbium gallium garnet) single crystal with a Verdet constant of approximately −134 rad·T$^{-1}$·m$^{-1}$ is the most widely used magneto-optical material [5–7]. However, owing to the size limitation of the crystal material and the relatively low Verdet constant, sesquioxide materials with higher Verdet constants, such as holmium oxide, terbium oxide, etc. [8–10], have attracted considerable attention. Among the sesquioxides, Tb$_2$O$_3$ has many excellent properties, such as high angular momentum [11] and only one absorption peak at 483 nm [12]. However, terbium sesquioxide easily oxidized into Tb$_4$O$_7$ without magneto-optical properties, which considerably affect the performance of this kind of material [13]. Furthermore, terbium oxide undergoes a reversible phase transition from the C-type cubic phase to the B-type monoclinic phase when the temperature exceeds 1600 °C. [14,15].

In order to prevent oxidation of terbium oxide, commercial Tb$_2$O$_3$ powders were used as original powder and sintered in an oxygen-free environment [16]. Some researchers attempted to deoxidize commercial Tb$_4$O$_7$ powder to Tb$_2$O$_3$ powder by hydrogen sintering [17] to reduce the sintering temperature. In the view of the phase change of terbium oxide during sintering, researchers have proposed a solution of doping rare earth elements [18–20]. Moreover, sintering additives including ZrO$_2$, La$_2$O$_3$, MgO, etc., were added to reduce the sintering temperature [21,22]. All these strategies represent promising techniques.

In 2017, Ning et al. [23] used low-level (0.5 wt%) $ZrO_2$-MgO as a double-sintering aid to prepare highly transparent $Yb:Y_2O_3$ ceramics. The addition of 0.5 wt% $ZrO_2$ was found to effectively increase the density of the ceramics and promote pore elimination. In 2019, Hu et al. [24] prepared $Dy_2O_3$ transparent ceramics by vacuum sintering of nanopowders. A $Dy_2O_3$ phase appeared at 600 °C during the decomposition period of the precursor, and the in-line transmittance values of the optimal ceramic sample with 1.0 mm thickness are 75.3% at 2000 nm and 67.9% at 633 nm. Despite many studies on $Yb_2O_3$, $Dy_2O_3$ and other sesquioxide materials, few studies have investigated the effect of sintering additives on the sintering properties of $Tb_2O_3$ ceramics.

In this work, $(Tb_{0.7}Lu_{0.3})_2O_3$ transparent ceramics were prepared by hydrogen pre-sintering combined with hot isostatic pressing sintering. We added varying amounts of $ZrO_2$ as sintering aids to promote ceramic densification. The effect of $ZrO_2$ content on the microstructure and sintering properties of these samples were investigated.

## 2. Experiment

### 2.1. Preparation Processes

High-purity $Tb_2O_3$ (99.99%), $Lu_2O_3$ (99.99%) and $ZrO_2$(99.99%) powders were used as raw materials. The raw materials were weighted according to the chemical formula of $(Tb_{0.7}Lu_{0.3})_2O_3$ and 0~5 at% $ZrO_2$ and mixed by ball milling for 24 h at a speed of 200 r/min. After drying by rotary evaporator, the dried powders were sieved through 100 meshes. The green ceramic samples were dry-pressed into a disk with a diameter of 20 mm at 20 MPa and further cold isostatically pressed at 200 MPa for 120s to increase the density. Before hydrogen sintering, the samples were calcined at 1000 °C for 10 h to remove organic impurities and residual carbon. The green bodies were sintered between 1550 °C and 1700 °C for 4 h in a hydrogen atmosphere. All the presintered samples were hot isostatically pressed at 1575 °C in an argon atmosphere to eliminate closed pores to improving performance of the ceramics. Finally, transparent $(Tb_{0.7}Lu_{0.3})_2O_3$ ceramics were obtained by mirror polishing to 2.0 mm thickness for measurement.

### 2.2. Characterization

The phase compositions of the ceramics were determined by X-ray diffraction (XRD; DX-1000CSC, Tongda Co. Ltd., Dandong, Liaoning, China) using Cu Kα radiation with a scan speed of 10°/min and a step size of 0.03° in the range of 2θ = 10°–70°. The linear shrinkage rate of the ceramics was calculated according to the following formula:

$$\frac{\Delta L}{L_0} = \frac{L_1 - L_0}{L_0} \tag{1}$$

where $\triangle L$ is the linear shrinkage of the sample after sintering, $L_0$ is the size of the green body before sintering and $L_1$ is the size of the samples after sintering. Microstructures of the fracture surfaces were examined using field emission scanning electron microscopy (SEM, Inspect F, FEI, Hillsborocity, OR, USA). The in-line transmittance of ceramics was measured by a UV-VIS-NIR spectrometer (Lambda950, PerkinElmer, Waltham, MA, USA) over the wavelength region from 200 nm to 1600 nm. Verdet constants were recorded on a Faraday effect experimental instrument (FD-FZ-C, Shanghai Fudan Tianxin Scientific & Educational Instruments Co., Ltd., Shanghai, China), with a laser wavelength of 650 nm.

## 3. Result and Discussion

Figure 1 presents XRD patterns of the $(Tb_{0.7}Lu_{0.3})_2O_3$ ceramic samples with varying $Zr^{4+}$ concentrations after hot isostatic pressing sintering. Compared with the standard XRD spectra of $Lu_2O_3$ (PDF#74-1980) and $Tb_2O_3$ (PDF#43-1012), there is no other impurity phase, and all the samples are approximately consistent with the characteristic peak of $Tb_2O_3$. The four diffraction peaks at 29°, 33.6°, 48.5° and 57.64° in the figure correspond to the crystal face of the cubic phases $Tb_2O_3$ (222), (400), (440) and (622), respectively [11]. The diffraction patterns located between the diffraction patterns of $Tb_2O_3$ and $Lu_2O_3$ indicate

that the $Lu^{3+}$ successfully entered the $Tb_2O_3$ lattice and replaced the $Tb^{3+}$ sites to form a $(Tb_{0.7}Lu_{0.3})_2O_3$ solid solution.

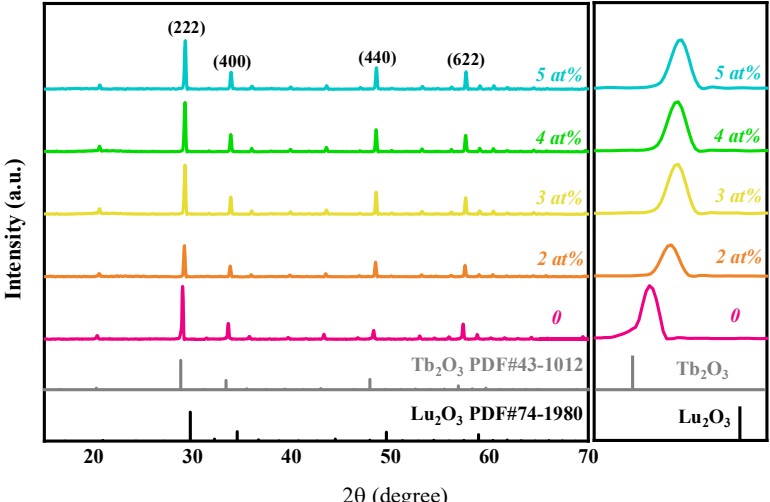

**Figure 1.** XRD patterns of $(Tb_{0.7}Lu_{0.3})_2O_3$ ceramics with 0~5 at% $ZrO_2$ as sintering aids after hot isostatic pressing sintering.

With the increase in $ZrO_2$ content, the characteristic peak shifts towards higher angles, which can be explained by the formulae (2) and (3), for which the standard notations of Kröger and Vink were used [25]. According to previous research [23], when $ZrO_2$ is added as a sintering aid, $Zr^{4+}$ replaces the position of $Tb^{3+}/Lu^{3+}$, and the following reactions occur to generate cation vacancies, which reduce the cell volume.

$$3ZrO_2 \overset{2Tb_2O_3}{\Longleftrightarrow} 3Zr_{Tb}^{\cdot} + V_{Tb}''' + 6O_o, \tag{2}$$

$$3ZrO_2 \overset{2Lu_2O_3}{\Longleftrightarrow} 3Zr_{Lu}^{\cdot} + V_{Lu}''' + 6O_o, \tag{3}$$

In addition, the radius of $Zr^{4+}$ (0.072 nm) is smaller than that of $Tb^{3+}$ (0.0923 nm) and $Lu^{3+}$ (0.0861 nm), which may also decrease the lattice cell constant.

Figure 2 shows the changes in lattice constant calculated by the XRD patterns. The cell size of the undoped $(Tb_{0.7}Lu_{0.3})_2O_3$ ceramic is approximately 10.4 angstrom. When zirconia is added from 2 at% to 3 at%, the cell size decreases from 10.37 Å to 10.35 Å. However, the lattice constant does not change linearly with the zirconia content. When the zirconia content exceeds 4 at%, the change in lattice constant is not obviously (approximately 0.01 Å), which could be ascribed to the zirconia solid solubility limit. The excess zirconia cannot enter the lattice, with an insignificant effect on the lattice constant. When the zirconia content is 5 at%, the lattice constant is 10.35 Å.

Figure 3 shows the change in linear shrinkage of $(Tb_{0.7}Lu_{0.3})_2O_3$ ceramics with varying $ZrO_2$ contents after hydrogen sintering from 1550 °C to 1700 °C. As shown in the figure, linear shrinkage increases with increased temperature. The shrinkage of the samples increases with increased zirconia content at the same sintering temperature. The linear shrinkage rate of the sample with no additional sintering additives increased linearly in the range of 1550 °C to 1700 °C. However, when $ZrO_2$ was added, a plateau gradually appeared around the temperature of 1600 °C~1650 °C, and the curve of those samples with zirconia as a sintering aid could be divided into two sections. Before 1600 °C, the sample shrank rapidly, and the curve slope was relatively high. When the temperature exceeded 1600 °C, the slope was reduced with increased $Zr^{4+}$ doping, proving that $ZrO_2$ can improve the densification rate and that ceramics can be rapidly densified below 1600 °C with sufficient zirconia content. The shrinkage curves were nearly identical when the doping content of $Zr^{4+}$ exceeded 3 at%.

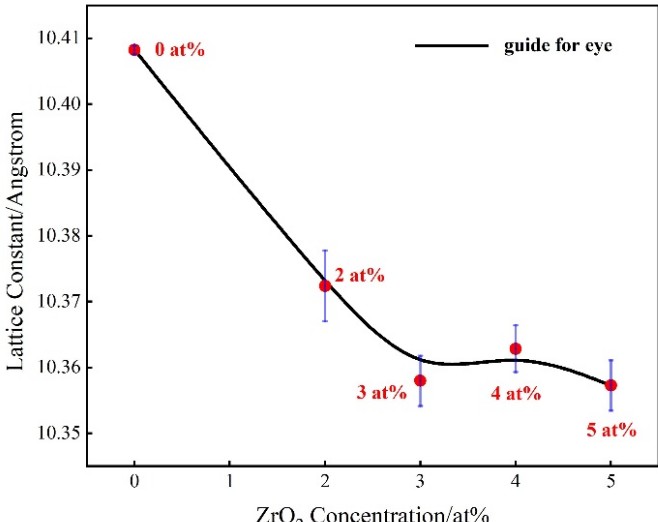

**Figure 2.** Change in lattice constants with varying Zr concentrations.

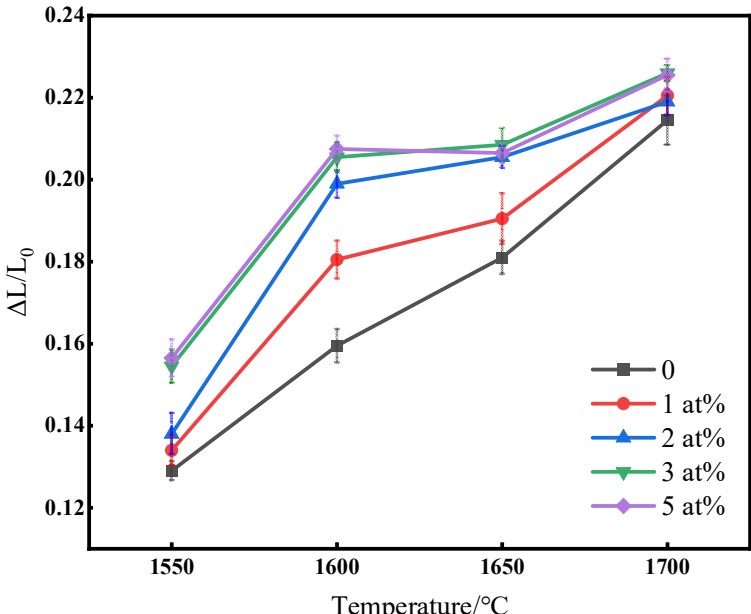

**Figure 3.** Linear shrinkage rate with varying $ZrO_2$ contents after hydrogen sintering from 1550 °C to 1700 °C.

According to the results presented above, we presintered the ceramics in a hydrogen atmosphere at 1550 °C, combined with hot isostatic pressing sintering at 1575 °C to make the samples fully dense. $Tb_2O_3$ undergoes severe phase transformation at 1600 °C [12,13], so a relatively low sintering temperature was chosen to avoid this transformation.

The microstructure of $(Tb_{0.7}Lu_{0.3})_2O_3$ ceramics after hydrogen sintering at 1550 °C with varying $ZrO_2$ contents is shown in Figure 4. With an increase in $Zr^{4+}$ content from 0 to 3 at%, the pores become smaller, suggesting that the addition of $ZrO_2$ can effectively reduce the sintering barrier of $(Tb_{0.7}Lu_{0.3})_2O_3$ ceramics and promote pore elimination during presintering. Subsequently, the phenomenon of pore shrinkage became indistinct when the addition amount increased to 4 at%. As shown in Figure 4e, the porosity and pore size increased considerably, suggesting that excessive zirconia is unfavorable for pore elimination during the presintering process.

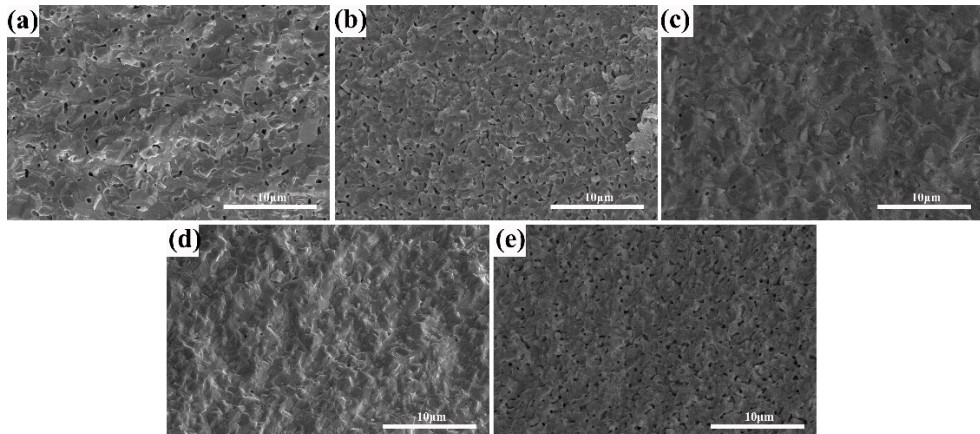

**Figure 4.** SEM photomicrographs of the fractured surfaces of the $(Tb_{0.7}Lu_{0.3})_2O_3$ ceramics after 1500 °C hydrogen sintering with varying $ZrO_2$ contents: (**a**) 0 at%, (**b**) 2 at%, (**c**) 3 at%, (**d**) 4 at%, (**e**) 5 at%.

Figure 5 shows SEM image of the ceramics with varying $ZrO_2$ contents after hot isostatic pressing sintering. The samples showed mainly transcrystalline fractures, and all samples presented with high density. As shown in Figure 5a, a large number of intragranular pores were observed, with a relatively large grain size.

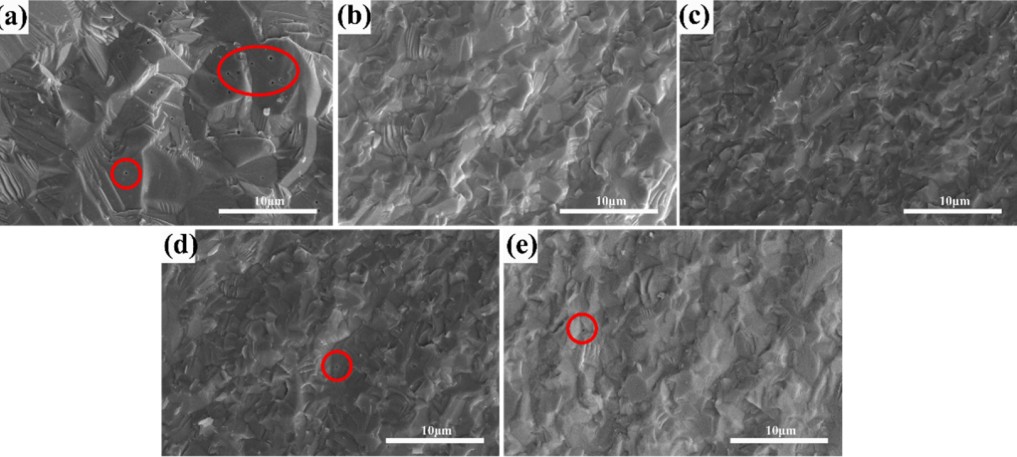

**Figure 5.** SEM micrographs of the fractured surfaces of the $(Tb_{0.7}Lu_{0.3})_2O_3$ ceramics after 1575 °C hot isostatic pressing sintering with varying $ZrO_2$ contents (The pores were marked in red circles): (**a**) 0 at%, (**b**) 2 at%, (**c**) 3 at%, (**d**) 4 at%, (**e**) 5 at%.

On the contrary, no obvious pores can be observed in Figure 5b,c. When Zr content exceeds 3 at%, some intragranular pores can be observed in Figure 5d,e. The grain size decreases from 2.23 μm to 1.92 μm with increased $ZrO_2$ content, which could be attributed to the doping effect of $ZrO_2$, which has been reported in many other sesquioxide ceramic sintering processes [26–29]. During the high-temperature sintering process of sesquioxide ceramics, the sintering aid, $ZrO_2$, can enter the lattice of sesquioxide as $Zr^{4+}$, as shown in Formulae (2) and (3). It can inhibit the rapid growth of ceramic grains, avoid the formation of intracrystalline pores and contribute to the full densification of ceramics. As shown in the figure, this effect is closely related to the concentration of zirconia. When the concentration is 3 at% or less, the densification of zirconia is considerable, but a further increase in the content can result in densifying hazards.

Figure 6 shows the in-line transmittance and appearance of $(Tb_{0.7}Lu_{0.3})_2O_3$ transparent ceramics with varying contents of $ZrO_2$ additive. Figure 6a shows that the samples sintered with $ZrO_2$ exhibited sufficient optical quality, so the words below the ceramics

could be clearly seen, and the optical quality was better than that of the ceramic sample without additive. This result is also confirmed by the transmittance curve of the sample shown in Figure 6b. The sample without sintering additives has the lowest transmittance, which could be related to the intracrystalline pores, as shown in Figure 5. With increased $Zr^{4+}$ content, the transparency first increased and then decreased, which is also related to the effect of zirconia. When zirconia content is excessive, it changes the sintering properties of the material, simultaneously affecting the internal microstructure (as shown in Figures 4 and 5), ultimately reducing the transparency. The ceramic sample with 3 at% $ZrO_2$ addition has the best optical quality, with a transparency of 74.96% at 1064 nm and 75.01% at 1550 nm. However, the transparency decreases with a further increase in $Zr^{4+}$ content to 4 at%, which could be related to the change in the internal microstructure of the ceramics. All the transmission curves show obvious emission peaks at 483 nm, in association with the $Tb^{3+}$ energy transition from $^7F_6$ to $^5D_4$ [11].

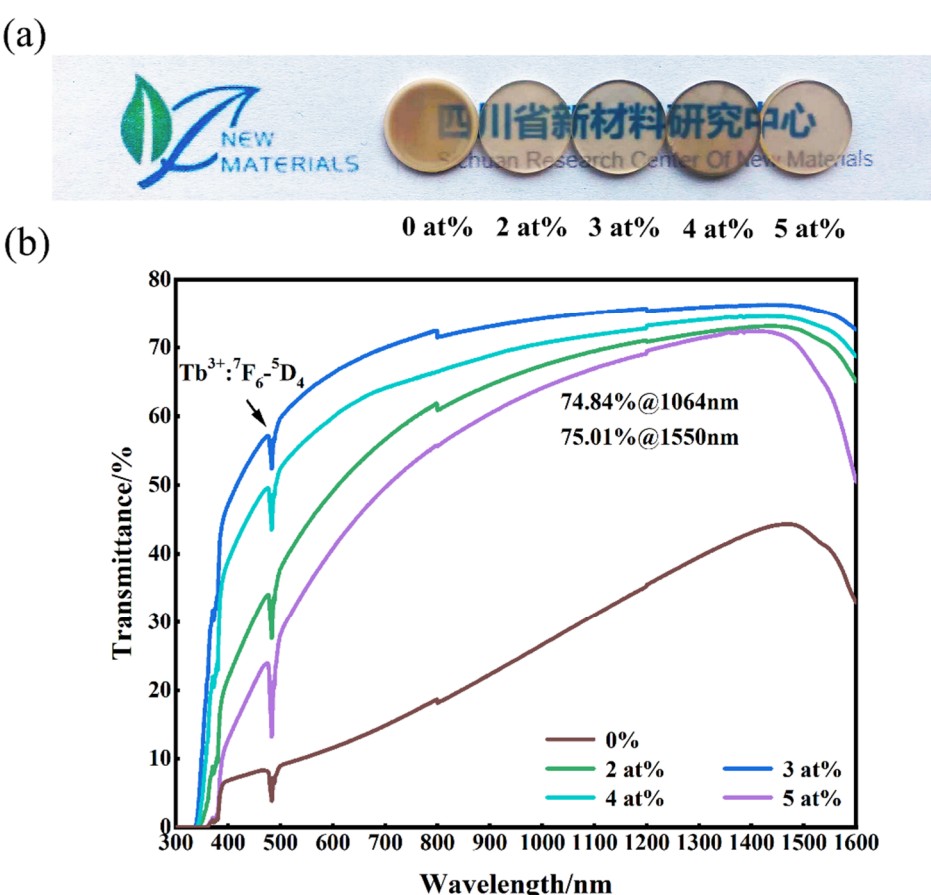

**Figure 6.** Photograph (**a**), and in-line transmittances curves (**b**) of the mirror–polished $(Tb_{0.7}Lu_{0.3})_2O_3$ ceramics sintered at 1550 °C followed by HIP at 1575 °C. (Zirconia content from left to right is 0 at%, 2 at%, 3 at%, 4 at% and 5 at%).

Figure 7a shows the microstructure of 3 at% $ZrO_2$ doped $(Tb_{0.7}Lu_{0.3})_2O_3$ ceramics. The fracture surface exhibited a pore-free structure, and the grains were tightly bound to each other. In addition, no secondary phase grain was observed on the grain boundary. As shown in Figure 5b, the particle size distribution of 3 at% Zr: $(Tb_{0.7}Lu_{0.3})_2O_3$ is mainly in the range of 1.0–2 μm, and the average grain size is approximately 1.74 μm.

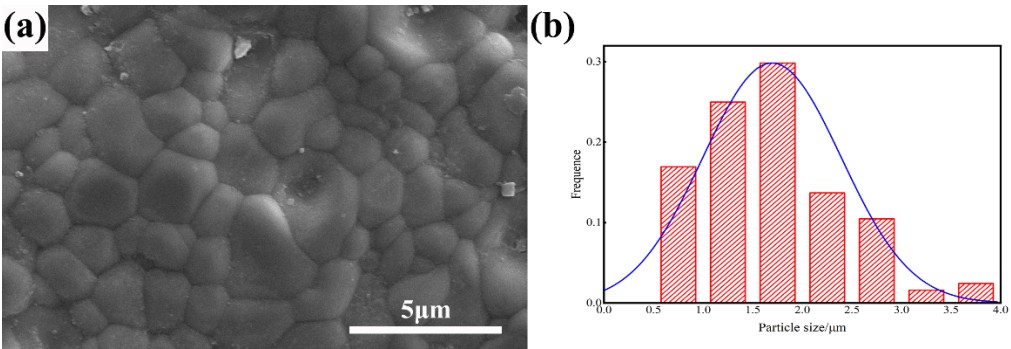

**Figure 7.** SEM micrographs of the surface of 3 at% Zr: $(Tb_{0.7}Lu_{0.3})_2O_3$ ceramic after hot isostatic pressing sintering (**a**) and the particle size distribution (**b**).

Figure 8 shows the Verdet constant of $(Tb_{0.7}Lu_{0.3})_2O_3$ ceramics with varying $Zr^{4+}$ contents at the 650 nm wavelength at room temperature. The Verdet constants for the ceramic samples were calculated using the following formula:

$$\theta = VHL \tag{4}$$

where $V$ is the Verdet constant, $\theta$ is the rotation angle of the light vector and $L$ is the length of the magneto-optical material. All samples were polished to 2 mm. As shown in the figure, the Vedet constant decreases from 339.36 rad·T$^{-1}$·m$^{-1}$ to 235.22 rad·T$^{-1}$·m$^{-1}$ with an increase in $Zr^{4+}$ content from 0 at% to 5 at%. The Verdet constants of $(Tb_{0.7}Lu_{0.3})_2O_3$ ceramics changed depending on the $Zr^{4+}$ concentration. According to the literature [30,31], for $Tb^{3+}$ ions, there are unpaired free electrons on the 4f electron layer, which produce an uncompensated magnetic moment in the magnetic field, which is the source of the magnetism. In the external magnetic field, electrons are prone to $^4f_8-^4f_7^5d$ transitions, which correspond to $^7F_6-^7D_5$ level transition, demonstrating strong magnetism.

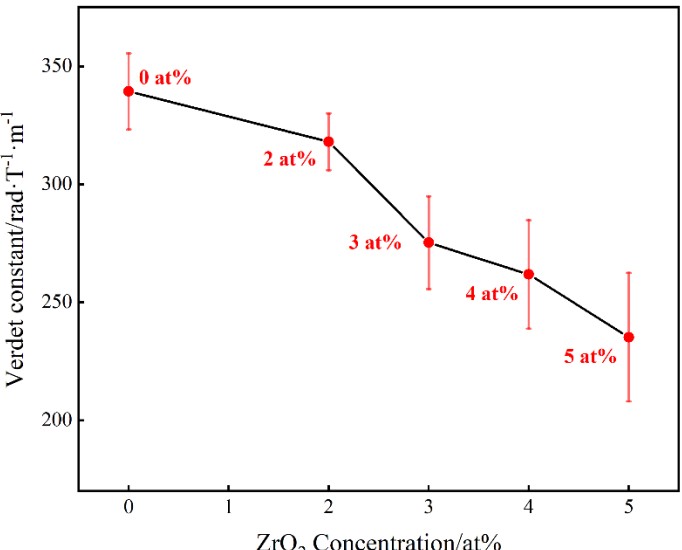

**Figure 8.** Verdet constant of Zr: $(Tb_{0.7}Lu_{0.3})_2O_3$ ceramics versus $Zr^{4+}$ concentration at the 650 nm wavelength at room temperature.

This phenomenon can be explained as follows. A large number of lattice vacancies and defects are produced with the addition of $ZrO_2$, which may affect the electron transition of $Tb^{3+}$, with an eventual decrease in the Verdet constant. Although the Verdet constant decreases with increased $Zr^{4+}$ content, the Verdet constant of $(Tb_{0.7}Lu_{0.3})_2O_3$ ceramics with

3 at% $ZrO_2$ is 275.28 rad·T$^{-1}$·m$^{-1}$, which is still about 2.05 times that of a commercial TGG single crystal.

## 4. Conclusions

Highly transparent $(Tb_{0.7}Lu_{0.3})_2O_3$ ceramics were prepared using $ZrO_2$ as a sintering additive a via hot isostatic pressing sintering process. With increased $Zr^{4+}$ content, the grain size decreases; the transparency of the sample increases first and then decreases. The inline transmittance of 3 at% $ZrO_2$ doped $(Tb_{0.7}Lu_{0.3})_2O_3$ ceramics reached 74.84% at 1064 nm. The addition of $ZrO_2$ can effectively reduce the sintering barrier of $(Tb_{0.7}Lu_{0.3})_2O_3$ ceramics, inhibit the abnormal growth of grains and promote the discharge of pores. The average grain size of the ceramics with 3 at% $ZrO_2$ added is approximately 1.74 μm. The Verdet constant of 3 at% $ZrO_2$-doped ceramics is 275.28 rad·T$^{-1}$·m$^{-1}$, which is still approximately 2.05 times that of a commercial TGG single crystal.

**Author Contributions:** Conceptualization, W.J.; Methodology, Y.X., T.X. and W.J.; Data analysis, Y.X., Y.W. and P.L.; Resources, B.K., W.J. and B.M.; Supervision, B.K. and W.J.; Writing—original draft, Y.X.; Writing—review and editing, W.J. and W.L. All authors have read and agreed to the published version of the manuscript.

**Funding:** This work was supported by the Development Foundation of China Academy of Engineering Physics and the National Natural Science Foundation of China (Grant No. 51902234).

**Institutional Review Board Statement:** Not applicable.

**Informed Consent Statement:** Not applicable.

**Data Availability Statement:** The data presented in this study are available upon request from the corresponding authors.

**Conflicts of Interest:** The authors declare no conflict of interest.

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
