# Peer review of "Effect of ZrO2 Content on Microstructure Evolution and Sintering Properties of (Tb0.7Lu0.3)2O3 Magneto-Optic Transparent Ceramics"

_magnetochemistry, doi:10.3390/magnetochemistry8120175_

Round 1

Reviewer 1 Report

This manuscript by Xin et al. reports on the sintering of Tb/Lu oxide in the presence of Zr oxide as a sintering aid. They found that ZrO2 improves the optical quality of the sintered samples and determined its optimal concentration. The work presents a rather extensive characterization and is in general of good quality. There are however some points that should be addressed and/or expanded.

Here are the major points:

- The authors could try estimating the average crystallite size of each sample by performing Scherrer analysis on peak width in XRD patterns. This could be used as a quantitative indicator of the sample microstructure.

- I fail to see appreciable differences between SEM micrographs of the Zr-doped samples. Both in-plane pictures from figure 4 and sections from figure 5 seem extremely similar, with the exception of the undoped sample in panels a. Please modify the discussion or present different pictures.

- In transmittance measurements, the wavelength 1064 nm is used as a reference. I wonder why the authors did not choose the main telecom wavelength of 1550 nm, which is currently by far the most interesting one.

- The authors should mention the wavelength at which the Verdet constants were evaluated. Moreover, they should add details on the experimental setup used to acquire magneto-optical data and the treatment to extract Verdet constants.

- When discussing magneto-optical performance, a Figure of Merit form is more significant than the plain Verdet constant. A common FoM for optical isolator materials is the ratio between unitary rotation and unitary attenuation (usually in dB). Also the comparison with TGG should be done in these terms, as well as specifying the working wavelength.

Other minor points:

- Line 26: please define TGG.

- Lines 80-81: a drying step is mentioned, but the text above does not mention the addition of a solvent. Please discuss/correct.

- Figure 2: please specify the nature of the black line (e.g.: guide for the eye).

- Line 192: please change ‘emission’ to ‘absorption’.

- Line 209: please change ‘Figure 5’ to ‘Figure 7’.

Reviewer 2 Report

The authors demonstrate the influence of ZrO2 content during the sintering of (Tb0.7Lu0.3)2O3 ceramics on its properties including magneto-optical effects. The results are compared with other materials including a widely used TGG. Optimal concentration of ZrO2 is shown in terms of transparency and Verdet constant.

I recommend this paper to be accepted after minor revision:

- The article could benefit from showing the error bars for the data in figures 2, 3, and 8.

- Line 174 of text looks broken.

- For the fig. 8 the blue dot doesn't correspond to 6%at concentration of ZrO2. I'd recommend showing the Verdet constant for TGG in a different way.
